# Individual Planning in Infinite-Horizon Multiagent Settings: Inference, Structure and Scalability

**Xia Qu**
Epic Systems
Verona, WI 53593
quxiapisces@gmail.com

**Prashant Doshi**
THINC Lab, Dept. of Computer Science
University of Georgia, Athens, GA 30622
pdoshi@cs.uga.edu

## Abstract

This paper provides the first formalization of self-interested planning in multiagent settings using expectation-maximization (EM). Our formalization in the context of infinite-horizon and finitely-nested interactive POMDPs (I-POMDP) is distinct from EM formulations for POMDPs and cooperative multiagent planning frameworks. We exploit the graphical model structure specific to I-POMDPs, and present a new approach based on block-coordinate descent for further speed up. Forward filtering-backward sampling – a combination of exact filtering with sampling – is explored to exploit problem structure.

## 1   Introduction

Generalization of bounded policy iteration (BPI) to finitely-nested interactive partially observable Markov decision processes (I-POMDP) [1] is currently the leading method for *infinite-horizon* self-interested multiagent planning and obtaining finite-state controllers as solutions. However, interactive BPI is acutely prone to converge to local optima, which severely limits the quality of its solutions despite the limited ability to escape from these local optima.

Attias [2] posed planning using MDP as a likelihood maximization problem where the "data" is the initial state and the final goal state or the maximum total reward. Toussaint et al. [3] extended this to infer finite-state automata for infinite-horizon POMDPs. Experiments reveal good quality controllers of small sizes although run time is a concern. Given BPI's limitations and the compelling potential of this approach in bringing advances in inferencing to bear on planning, we generalize it to infinite-horizon and finitely-nested I-POMDPs. Our generalization allows its use toward planning for an individual agent in noncooperation where we may not assume common knowledge of initial beliefs or common rewards, due to which others' beliefs, capabilities and preferences are modeled.

Analogously to POMDPs, we formulate a mixture of finite-horizon DBNs. However, the DBNs differ by including models of other agents in a special model node. *Our approach, labeled as I-EM, improves on the straightforward extension of Toussaint et al.'s EM to I-POMDPs by utilizing various types of structure.* Instead of ascribing as many level 0 finite-state controllers as candidate models and improving each using its own EM, we use the underlying graphical structure of the model node and its update to formulate a single EM that directly provides the marginal of others' actions across all models. This rests on a new insight, which considerably simplifies and speeds EM at level 1.

We present a general approach based on block-coordinate descent [4, 5] for speeding up the non-asymptotic rate of convergence of the iterative EM. The problem is decomposed into optimization subproblems in which the objective function is optimized with respect to a small subset (block) of variables, while holding other variables fixed. We discuss the unique challenges and present the first effective application of this iterative scheme to multiagent planning.

Finally, sampling offers a way to exploit the embedded problem structure such as information in distributions. The exact forward-backward E-step is replaced with *forward filtering-backward sampling*

*(FFBS)* that generates trajectories weighted with rewards, which are used to update the parameters of the controller. While sampling has been integrated in EM previously [6], FFBS specifically mitigates error accumulation over long horizons due to the exact forward step.

## 2   Overview of Interactive POMDPs

A finitely-nested I-POMDP [7] for an agent $i$ with strategy level, $l$, interacting with agent $j$ is: I-POMDP$_{i,l} = \langle IS_{i,l}, A, T_i, \Omega_i, O_i, R_i, OC_i \rangle$

- $IS_{i,l}$ denotes the set of *interactive states* defined as, $IS_{i,l} = S \times M_{j,l-1}$, where $M_{j,l-1} = \{\Theta_{j,l-1} \cup SM_j\}$, for $l \geq 1$, and $IS_{i,0} = S$, where $S$ is the set of physical states. $\Theta_{j,l-1}$ is the set of computable, intentional models ascribed to agent $j$: $\theta_{j,l-1} = \langle b_{j,l-1}, \hat{\theta}_j \rangle$. Here $b_{j,l-1}$ is agent $j$'s level $l-1$ belief, $b_{j,l-1} \in \triangle(IS_{j,l-1})$ where $\Delta(\cdot)$ is the space of distributions, and $\hat{\theta}_j = \langle A, T_j, \Omega_j, O_j, R_j, OC_j \rangle$, is $j$'s *frame*. At *level l=0*, $b_{j,0} \in \triangle(S)$ and a intentional model reduces to a POMDP. $SM_j$ is the set of subintentional models of $j$, an example is a finite state automaton.
- $A = A_i \times A_j$ is the set of joint actions of all agents.
- Other parameters – transition function, $T_i$, observations, $\Omega_i$, observation function, $O_i$, and preference function, $R_i$ – have their usual semantics analogously to POMDPs but involve joint actions.
- Optimality criterion, $OC_i$, here is the discounted infinite horizon sum.

An agent's belief over its interactive states is a sufficient statistic fully summarizing the agent's observation history. Given the associated belief update, solution to an I-POMDP is a policy. Using the Bellman equation, each belief state in an I-POMDP has a value which is the maximum payoff the agent can expect starting from that belief and over the future.

## 3   Planning in I-POMDP as Inference

We may represent the policy of agent $i$ for the infinite horizon case as a stochastic *finite state controller (FSC)*, defined as: $\pi_i = \langle \mathcal{N}_i, \mathcal{T}_i, \mathcal{L}_i, \mathcal{V}_i \rangle$ where $\mathcal{N}_i$ is the set of nodes in the controller. $\mathcal{T}_i : \mathcal{N}_i \times A_i \times \Omega_i \times \mathcal{N}_i \to [0,1]$ represents the node transition function; $\mathcal{L}_i : \mathcal{N}_i \times A_i \to [0,1]$ denotes agent $i$'s action distribution at each node; and an initial distribution over the nodes is denoted by, $\mathcal{V}_i : \mathcal{N}_i \to [0,1]$. For convenience, we group $\mathcal{V}_i, \mathcal{T}_i$ and $\mathcal{L}_i$ in $\hat{f}_i$. Define a controller at level $l$ for agent $i$ as, $\pi_{i,l} = \langle \mathcal{N}_{i,l}, \hat{f}_{i,l} \rangle$, where $\mathcal{N}_{i,l}$ is the set of nodes in the controller and $\hat{f}_{i,l}$ groups remaining parameters of the controller as mentioned before. Analogously to POMDPs [3], we formulate planning in multiagent settings formalized by I-POMDPs as a likelihood maximization problem:

$$\pi_{i,l}^* = \arg\max_{\Pi_{i,l}} (1-\gamma) \sum_{T=0}^{\infty} \gamma^T \, Pr(r_i^T = 1 | T; \pi_{i,l}) \tag{1}$$

where $\Pi_{i,l}$ are all level-$l$ FSCs of agent $i$, $r_i^T$ is a binary random variable whose value is 0 or 1 emitted after $T$ time steps with probability proportional to the reward, $R_i(s, a_i, a_j)$.

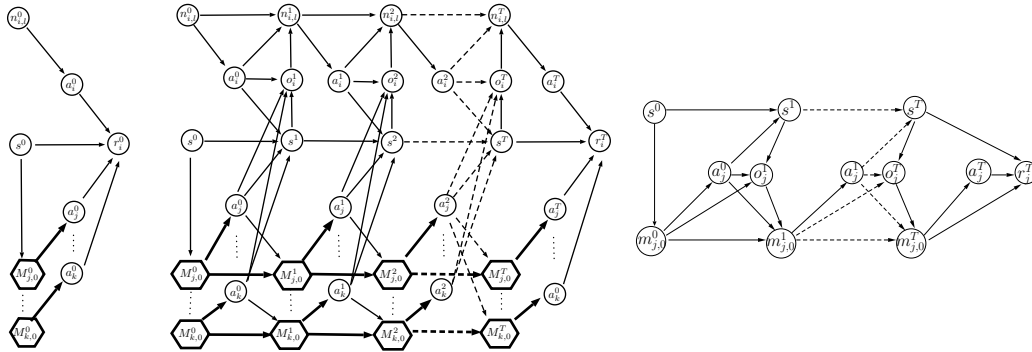

Figure 1: $(a)$ Mixture of DBNs with 1 to $T$ time slices for I-POMDP$_{i,1}$ with $i$'s level-1 policy represented as a standard FSC whose "node state" is denoted by $n_{i,l}$. The DBNs differ from those for POMDPs by containing special *model nodes* (hexagons) whose values are candidate models of other agents. $(b)$ Hexagonal model nodes and edges in bold for one other agent $j$ in $(a)$ decompose into this level-0 DBN. Values of the node $m_{j,0}^t$ are the candidate models. CPT of chance node $a_j^t$ denoted by $\phi_{j,0}(m_{j,0}^t, a_j^t)$ is inferred using likelihood maximization.

The planning problem is modeled as a mixture of DBNs of increasing time from $T=0$ onwards (Fig. 1). The transition and observation functions of I-POMDP$_{i,l}$ parameterize the chance nodes $s$ and $o_i$, respectively, along with $Pr(r_i^T|a_i^T, a_j^T, s^T) \propto \frac{R_i(s^T, a_i^T, a_j^T) - R_{min}}{R_{max} - R_{min}}$. Here, $R_{max}$ and $R_{min}$ are the maximum and minimum reward values in $R_i$.

The networks include nodes, $n_{i,l}$, of agent $i$'s level-$l$ FSC. Therefore, functions in $\hat{f}_{i,l}$ parameterize the network as well, which are to be inferred. Additionally, the network includes the hexagonal *model nodes* – one for each other agent – that contain the candidate level 0 models of the agent. Each model node provides the expected distribution over another agent's actions. Without loss of generality, no edges exist between model nodes in the same time step. Correlations between agents could be included as state variables in the models.

Agent $j$'s model nodes and the edges (in bold) between them, and between the model and chance action nodes represent a DBN of length $T$ as shown in Fig. 1($b$). Values of the chance node, $m_{j,0}^0$, are the candidate models of agent $j$. Agent $i$'s initial belief over the state and models of $j$ becomes the parameters of $s^0$ and $m_{j,0}^0$. The likelihood maximization at level 0 seeks to obtain the distribution, $Pr(a_j|m_{j,0}^0)$, for each candidate model in node, $m_{j,0}^0$, using EM on the DBN.

**Proposition 1** (Correctness). *The likelihood maximization problem as defined in Eq. 1 with the mixture models as given in Fig. 1 is equivalent to the problem of solving the original I-POMDP$_{i,l}$ with discounted infinite horizon whose solution assumes the form of a finite state controller.*

All proofs are given in the supplement. Given the unique mixture models above, the challenge is to generalize the EM-based iterative maximization for POMDPs to the framework of I-POMDPs.

## 3.1 Single EM for Level 0 Models

The straightforward approach is to infer a likely FSC for each level 0 model. However, this approach does not scale to many models. Proposition 2 below shows that the *dynamic $Pr(a_j^t|s^t)$ is sufficient predictive information about other agent from its candidate models at time $t$, to obtain the most likely policy of agent $i$. This is markedly different from using behavioral equivalence [8] that clusters models with identical solutions. The latter continues to require the full solution of *each* model.

**Proposition 2** (Sufficiency). *Distributions $Pr(a_j^t|s^t)$ across actions $a_j^t \in A_j$ for each state $s^t$ is sufficient predictive information about other agent $j$ to obtain the most likely policy of $i$.*

In the context of Proposition 2, we seek to infer $Pr(a_j^t|m_{j,0}^t)$ for each (updated) model of $j$ at all time steps, which is denoted as $\phi_{j,0}$. Other terms in the computation of $Pr(a_j^t|s^t)$ are known parameters of the level 0 DBN. The likelihood maximization for the level 0 DBN is:

$$\phi_{j,0}^* = \arg \max_{\phi_{j,0}} (1 - \gamma) \sum_{T=0}^{\infty} \sum_{m_{j,0} \in M_{j,0}^T} \gamma^T Pr(r_j^T = 1|T, m_{j,0}; \phi_{j,0})$$

As the trajectory consisting of states, models, actions and observations of the other agent is hidden at planning time, we may solve the above likelihood maximization using EM.

**E-step** Let $z_j^{0:T} = \{s^t, m_{j,0}^t, a_j^t, o_j^t\}_0^T$ where the observation at $t=0$ is null, be the hidden trajectory. The log likelihood is obtained as an expectation of these hidden trajectories:

$$Q(\phi_{j,0}'|\phi_{j,0}) = \sum_{T=0}^{\infty} \sum_{z_j^{0:T}} Pr(r_j^T = 1, z_j^{0:T}, T; \phi_{j,0}) \, log \, Pr(r_j^T = 1, z_j^{0:T}, T; \phi_{j,0}') \qquad (2)$$

The "data" in the level 0 DBN consists of the initial belief over the state and models, $b_{i,1}^0$, and the observed reward at $T$. Analogously to EM for POMDPs, this motivates forward filtering-backward smoothing on a network with joint state $(s^t, m_{j,0}^t)$ for computing the log likelihood. The transition function for the forward and backward steps is:

$$Pr(s^t, m_{j,0}^t|s^{t-1}, m_{j,0}^{t-1}) = \sum_{a_j^{t-1}, o_j^t} \phi_{j,0}(m_{j,0}^{t-1}, a_j^{t-1}) \, T_{m_j}(s^{t-1}, a_j^{t-1}, s^t) \, Pr(m_{j,0}^t|m_{j,0}^{t-1}, a_j^{t-1}, o_j^t)$$
$$\times O_{m_j}(s^t, a_j^{t-1}, o_j^t) \qquad (3)$$

where $m_j$ in the subscripts is $j$'s model at $t-1$. Here, $Pr(m_{j,0}^t|a_j^{t-1}, o_j^t, m_{j,0}^{t-1})$ is the Kronecker-delta function that is 1 when $j$'s belief in $m_{j,0}^{t-1}$ updated using $a_j^{t-1}$ and $o_j^t$ equals the belief in $m_{j,0}^t$; otherwise 0.

Forward filtering gives the probability of the next state as follows:

$$\alpha^t(s^t, m_{j,0}^t) = \sum_{s^{t-1}, m_{j,0}^{t-1}} Pr(s^t, m_{j,0}^t | s^{t-1}, m_{j,0}^{t-1})\, \alpha^{t-1}(s^{t-1}, m_{j,0}^{t-1})$$

where $\alpha^0(s^0, m_{j,0}^0)$ is the initial belief of agent $i$. The smoothing by which we obtain the joint probability of the state and model at $t-1$ from the distribution at $t$ is:

$$\beta^h(s^{t-1}, m_{j,0}^{t-1}) = \sum_{s^t, m_{j,0}^t} Pr(s^t, m_{j,0}^t | s^{t-1}, m_{j,0}^{t-1})\, \beta^{h-1}(s^t, m_{j,0}^t)$$

where $h$ denotes the horizon to $T$ and $\beta^0(s^T, m_{j,0}^T) = E_{a_j^T | m_{j,0}^T}[Pr(r_j^T = 1 | s^T, m_{j,0}^T)]$. Messages $\alpha^t$ and $\beta^h$ give the probability of a state at some time slice in the DBN. As we consider a mixture of BNs, we seek probabilities for all states in the mixture model. Subsequently, we may compute the forward and backward messages at all states for the entire mixture model in one sweep.

$$\widehat{\alpha}(s, m_{j,0}) = \sum_{t=0}^{\infty} Pr(T = t)\, \alpha^t(s, m_{j,0}) \qquad \widehat{\beta}(s, m_{j,0}) = \sum_{h=0}^{\infty} Pr(T = h)\, \beta^h(s, m_{j,0}) \qquad (4)$$

**Model growth**   As the other agent performs its actions and makes observations, the space of $j$'s models grows exponentially: starting from a finite set of $|M_{j,0}^0|$ models, we obtain $\mathcal{O}(|M_{j,0}^0|(|A_j||\Omega_j|)^t)$ models at time $t$. This greatly increases the number of trajectories in $Z_j^{0:T}$. We limit the growth in the model space by sampling models at the next time step from the distribution, $\alpha^t(s^t, m_{j,0}^t)$, as we perform each step of forward filtering. It limits the growth by exploiting the structure present in $\phi_{j,0}$ and $O_j$, which guide how the models grow.

**M-step**   We obtain the updated $\phi_{j,0}'$ from the full log likelihood in Eq. 2 by separating the terms:

$$Q(\phi_{j,0}' | \phi_{j,0}) = \langle \text{terms independent of } \phi_{j,0}' \rangle + \sum_{T=0}^{\infty} \sum_{z_j^{0:T}} Pr(r_i^T = 1, z_j^{0:T}, T; \phi_{j,0}') \sum_{t=0}^{T} \phi_{j,0}'(a_j^t | m_{j,0}^t)$$

and maximizing it w.r.t. $\phi_{j,0}'$:

$$\phi_{j,0}'(a_j^t, m_{j,0}^t) \propto \phi_{j,0}(a_j^t, m_j^t) \sum_{s^t} R_{m_j}(s^t, a_j^t)\, \widehat{\alpha}(s^t, m_{j,0}^t) + \sum_{s^t, s^{t+1}, m_{j,0}^{t+1}, o_j^{t+1}} \frac{\gamma}{1-\gamma} \widehat{\beta}(s^{t+1}, m_{j,0}^{t+1})$$

$$\times\ \widehat{\alpha}(s^t, m_{j,0}^t)\, T_{m_j}(s^t, a_j^t, s^{t+1})\, Pr(m_{j,0}^{t+1} | m_{j,0}^t, a_j^t, o_j^{t+1})\, O_{m_j}(s^{t+1}, a_j^t, o_j^{t+1})$$

### 3.2   Improved EM for Level $l$ I-POMDP

At strategy levels $l \geq 1$, Eq. 1 defines the likelihood maximization problem, which is iteratively solved using EM. We show the $E$- and $M$-steps next beginning with $l = 1$.

**E-step**   In a multiagent setting, the hidden variables additionally include what the other agent may observe and how it acts over time. However, a key insight is that Prop. 2 allows us to limit attention to the marginal distribution over other agents' actions given the state. Thus, let $z_i^{0:T} = \{s^t, o_i^t, n_{i,l}^t, a_i^t, a_j^t, \ldots, a_k^t\}_0^T$, where the observation at $t = 0$ is null, and other agents are labeled $j$ to $k$; this group is denoted $-i$. The full log likelihood involves an expectation over hidden variables:

$$Q(\pi_{i,l}' | \pi_{i,l}) = \sum_{T=0}^{\infty} \sum_{z_i^{0:T}} Pr(r_i^T = 1, z_i^{0:T}, T; \pi_{i,l}) \, \log Pr(r_i^T = 1, z_i^{0:T}, T; \pi_{i,l}') \qquad (5)$$

Due to the subjective perspective in I-POMDPs, $Q$ computes the likelihood of agent $i$'s FSC only (and not of joint FSCs as in team planning [9]).

In the $T$-step DBN of Fig. 1, observed evidence includes the reward, $r_i^T$, at the end and the initial belief. We seek the likely distributions, $\mathcal{V}_i$, $\mathcal{T}_i$, and $\mathcal{L}_i$, across time slices. We may again realize the full joint in the expectation using a forward-backward algorithm on a hidden Markov model whose state is $(s^t, n_{i,l}^t)$. The transition function of this model is,

$$Pr(s^t, n_{i,l}^t | s^{t-1}, n_{i,l}^{t-1}) = \sum_{a_i^{t-1}, a_{-i}^{t-1}, o_i^t} \mathcal{L}_i(n_{i,l}^{t-1}, a_i^{t-1}) \prod_{-i} Pr(a_{-i}^{t-1} | s^{t-1})\, \mathcal{T}_i(n_{i,l}^{t-1}, a_i^{t-1}, o_i^t, n_{i,l}^t)$$

$$\times\ T_i(s^{t-1}, a_i^{t-1}, a_{-i}^{t-1}, s^t)\, O_i(s^t, a_i^{t-1}, a_{-i}^{t-1}, o_i^t) \qquad (6)$$

In addition to parameters of I-POMDP$_{i,l}$, which are given, parameters of agent $i$'s controller and those relating to other agents' predicted actions, $\phi_{-i,0}$, are present in Eq. 6. Notice that in consequence of Proposition 2, Eq. 6 precludes $j$'s observation and node transition functions.

The forward message, $\alpha^t = Pr(s^t, n_{i,l}^t)$, represents the probability of being at some state of the DBN at time $t$:

$$\alpha^t(s^t, n_{i,l}^t) = \sum\nolimits_{s^{t-1}, n_{i,l}^{t-1}} Pr(s^t, n_{i,l}^t | s^{t-1}, n_{i,l}^{t-1}) \, \alpha^{t-1}(s^{t-1}, n_{i,l}^{t-1}) \tag{7}$$

where, $\alpha^0(s^0, n_{i,l}^0) = \mathcal{V}_i(n_{i,l}^0) b_{i,l}^0(s^0)$. The backward message gives the probability of observing the reward in the final $T^{th}$ time step given a state of the Markov model, $\beta^t(s^t, n_{i,l}^t) = Pr(r_i^T = 1 | s^t, n_{i,l}^t)$:

$$\beta^h(s^t, n_{i,l}^t) = \sum\nolimits_{s^{t+1}, n_{i,l}^{t+1}} Pr(s^{t+1}, n_{i,l}^{t+1} | s^t, n_{i,l}^t) \, \beta^{h-1}(s^{t+1}, n_{i,l}^{t+1}) \tag{8}$$

where, $\beta^0(s^T, n_{i,l}^T) = \sum_{a_i^T, a_{-i}^T} Pr(r_i^T = 1 | s^T, a_i^T, a_{-i}^T) \, \mathcal{L}_i(n_{i,l}^T, a_i^T) \prod_{-i} Pr(a_{-i}^T | s^T)$, and $1 \leq h \leq T$ is the horizon. Here, $Pr(r_i^T = 1 | s^T, a_i^T, a_{-i}^T) \propto R_i(s^T, a_i^T, a_{-i}^T)$.

A side effect of $Pr(a_{-i}^t | s^t)$ being dependent on $t$ is that we can no longer conveniently define $\widehat{\alpha}$ and $\widehat{\beta}$ for use in $M$-step at level 1. Instead, the computations are folded in the $M$-step.

**M-step** We update the parameters, $\mathcal{L}_i$, $\mathcal{T}_i$ and $\mathcal{V}_i$, of $\pi_{i,l}$ to obtain $\pi'_{i,l}$ based on the expectation in the E-step. Specifically, take log of the likelihood $Pr(r^T = 1, z_i^{0:T}, T; \pi_{i,l})$ with $\pi_{i,l}$ substituted with $\pi'_{i,l}$ and focus on terms involving the parameters of $\pi'_{i,l}$:

$$\log Pr(r^T = 1, z_i^{0:T}, T; \pi'_{i,l}) = \langle \text{terms independent of } \pi'_{i,l} \rangle + \sum\nolimits_{t=0}^{T} \log \mathcal{L}'_i(n_{i,l}^t, a_i^t) +$$

$$\sum\nolimits_{t=0}^{T-1} \log \mathcal{T}'_i(n_{i,l}^t, a_i^t, o_i^{t+1}, n_{i,l}^{t+1}) + \log \mathcal{V}'_i(n_{i,l})$$

In order to update, $\mathcal{L}_i$, we partially differentiate the Q-function of Eq. 5 with respect to $\mathcal{L}'_i$. To facilitate differentiation, we focus on the terms involving $\mathcal{L}_i$, as shown below.

$$Q(\pi'_{i,l} | \pi_{i,l}) = \langle \text{terms indep. of } \mathcal{L}'_i \rangle + \sum\nolimits_{T=0}^{\infty} \Pr(T) \sum\nolimits_{t=0}^{T} \sum\nolimits_{z_i^{0:t}} \Pr(r_i^T = 1, z_i^{0:t} | T; \pi_{i,l}) \log \mathcal{L}'_i(n_{i,l}^t, a_i^t)$$

$\mathcal{L}'_i$ on maximizing the above equation is:

$$\mathcal{L}'_i(n_{i,l}^t, a_i^t) \propto \mathcal{L}_i(n_{i,l}^t, a_i^t) \sum\nolimits_{T=0}^{\infty} \prod\nolimits_{-i} \sum\nolimits_{s^T, a_{-i}^T} \frac{\gamma^T}{1 - \gamma} Pr(r_i^T = 1 | s^T, a_i^T, a_{-i}^T) \, Pr(a_{-i}^T | s^T) \, \alpha^T(s^T, n_{i,l}^T)$$

Node transition probabilities $\mathcal{T}_i$ and node distribution $\mathcal{V}_i$ for $\pi'_{i,l}$, is updated analogously to $\mathcal{L}_i$.

Because a FSC is inferred at level 1, at strategy levels $l = 2$ and greater, lower-level candidate models are FSCs. EM at these higher levels proceeds by replacing the state of the DBN, $(s^t, n_{i,l}^t)$ with $(s^t, n_{i,l}^t, n_{j,l-1}^t, \ldots, n_{k,l-1}^t)$.

### 3.3 Block-Coordinate Descent for Non-Asymptotic Speed Up

Block-coordinate descent (BCD) [4, 5, 10] is an iterative scheme to gain faster non-asymptotic rate of convergence in the context of large-scale $N$-dimensional optimization problems. In this scheme, within each iteration, a set of variables referred to as coordinates are chosen and the objective function, $Q$, is optimized with respect to one of the coordinate blocks while the other coordinates are held *fixed*. BCD may speed up the non-asymptotic rate of convergence of EM for both I-POMDPs and POMDPs. *The specific challenge here is to determine which of the many variables should be grouped into blocks and how.*

We empirically show in Section 5 that grouping the number of time slices, $t$, and horizon, $h$, in Eqs. 7 and 8, respectively, at each level, into coordinate blocks of equal size is beneficial. In other words, we decompose the mixture model into blocks containing equal numbers of BNs. Alternately, grouping controller nodes is ineffective because distribution $\mathcal{V}_i$ cannot be optimized for subsets of nodes. Formally, let $\Psi_1^t$ be a subset of $\{T = 1, T = 2, \ldots, T = T_{max}\}$. Then, the set of blocks is, $B_t = \{\Psi_1^t, \Psi_2^t, \Psi_3^t, \ldots\}$. In practice, because both $t$ and $h$ are finite (say, $T_{max}$), the cardinality of $B_t$ is bounded by some $C \geq 1$. Analogously, we define the set of blocks of $h$, denoted by $B_h$.

In the $M$-step now, we compute $\alpha^t$ for the time steps in a single coordinate block $\Psi_c^t$ only, while using the values of $\alpha^t$ from the *previous* iteration for the complementary coordinate blocks, $\tilde{\Psi}_c^t$. Analogously, we compute $\beta^h$ for the horizons in $\Psi_c^h$ only, while using $\beta$ values from the previous iteration for the remaining horizons. We cyclically choose a block, $\Psi_c^t$, at iterations $c + qC$ where $q \in \{0, 1, 2, \ldots\}$.

### 3.4 Forward Filtering - Backward Sampling

An approach for exploiting embedded structure in the transition and observation functions is to replace the exact forward-backward message computations with exact forward filtering and backward sampling (FFBS) [11] to obtain a sampled reverse trajectory consisting of $\langle s^T, n_{i,l}^T, a_i^T \rangle$, $\langle n_{i,l}^{T-1}, a_i^{T-1}, o_i^T, n_{i,l}^T \rangle$, and so on from T to 0. Here, $Pr(r_i^T = 1 | s^T, a_i^T, a_{-i}^T)$ is the likelihood weight of this trajectory sample. Parameters of the updated FSC, $\pi'_{i,l}$, are obtained by summing and normalizing the weights.

Each trajectory is obtained by first sampling $\hat{T} \sim Pr(T)$, which becomes the length of $i$'s DBN for this sample. Forward message, $\alpha^t(s^t, n_{i,l}^t)$, $t = 0 \ldots \hat{T}$ is computed exactly (Eq. 7) followed by the backward message, $\beta^h(s^t, n_{i,l}^t)$, $h = 0 \ldots \hat{T}$ and $t = \hat{T} - h$. *Computing $\beta^h$ differs from Eq. 8 by utilizing the forward message*:

$$\beta^h(s^t, n_{i,l}^t | s^{t+1}, n_{i,l}^{t+1}) = \sum_{a_i^t, a_{-i}^t, o_i^{t+1}} \alpha^t(s^t, n_{i,l}^t)\, \mathcal{L}_i(n_{i,l}^t, a_i^t) \prod_{-i} Pr(a_{-i}^t | s^t)\, T_i(s^t, a_i^t, a_{-i}^t, s^{t+1})$$
$$\mathcal{T}_i(n_{i,l}^t, a_i^t, o_i^{t+1}, n_{i,l}^{t+1})\, O_i(s^{t+1}, a_i^t, a_{-i}^t, o_i^{t+1}) \tag{9}$$

where $\beta^0(s^T, n_{i,l}^T, r_i^T) = \sum_{a_i^t, a_{-i}^t} \alpha^T(s^T, n_{i,l}^T) \prod_{-i} Pr(a_{-i}^T | s^T)\, \mathcal{L}(n_{i,l}^T, a_i^T)\, Pr(r_i^T | s^T, a_i^T, a_{-i}^T)$.
Subsequently, we may easily sample $\langle s^T, n_{i,l}^T, r_i^T \rangle$ followed by sampling $s_i^{T-1}, n_{i,l}^{T-1}$ from Eq. 9.

We sample $a_i^{T-1}, o_i^T \sim Pr(a_i^t, o_i^{t+1} | s^t, n_{i,l}^t, s^{t+1}, n_{i,l}^{t+1})$, where:

$$Pr(a_i^t, o_i^{t+1} | s^t, n_{i,l}^t, s^{t+1}, n_{i,l}^{t+1}) \propto \prod_{-i} Pr(a_{-i}^t | s^t)\, \mathcal{L}_i(n_{i,l}^t, a_i^t)\, \mathcal{T}_i(n_{i,l}^t, a_i^t, o_i^{t+1}, n_{i,l}^{t+1})\, T_i(s^t, a_i^t, a_{-i}^t, s^{t+1})$$
$$O_i(s^{t+1}, a_i^t, a_{-j}^t, o_i^{t+1})$$

## 4 Computational Complexity

Our EM at level 1 is significantly quicker compared to ascribing FSCs to other agents. In the latter, nodes of others' controllers must be included alongside $s$ and $n_{i,l}$.

**Proposition 3** (E-step speed up). *Each E-step at level 1 using the forward-backward pass as shown previously results in a net speed up of $\mathcal{O}((|M||\mathcal{N}_{-i,0}|)^{2K} |\Omega_{-i}|^K)$ over the formulation that ascribes $|M|$ FSCs each to K other agents with each having $|\mathcal{N}_{-i,0}|$ nodes.*

Analogously, updating the parameters $\mathcal{L}_i$ and $\mathcal{T}_i$ in the M-step exhibits a speedup of $\mathcal{O}((|M||\mathcal{N}_{-i,0}|)^{2K} |\Omega_{-i}|^K)$, while $\mathcal{V}_i$ leads to $\mathcal{O}((|M||\mathcal{N}_{-i,0}|)^K)$. This improvement is exponential in the number of other agents. On the other hand, the E-step at level 0 exhibits complexity that is typically greater compared to the total complexity of the E-steps for $|M|$ FSCs.

**Proposition 4** (E-step ratio at level 0). *E-steps when $|M|$ FSCs are inferred for K agents exhibit a ratio of complexity, $\mathcal{O}(\frac{|\mathcal{N}_{-i,0}|^2}{|M|})$, compared to the E-step for obtaining $\phi_{-i,0}$.*

The ratio in Prop. 4 is $< 1$ when smaller-sized controllers are sought and there are several models.

## 5 Experiments

Five variants of EM are evaluated as appropriate: the exact EM inference-based planning (labeled as I-EM); replacing the exact M-step with its greedy variant analogously to the greedy maximization in EM for POMDPs [12] (I-EM-Greedy); iterating EM based on coordinate blocks (I-EM-BCD) and coupled with a greedy M-step (I-EM-BCD-Greedy); and lastly, using forward filtering-backward sampling (I-EM-FFBS).

We use 4 problem domains: the noncooperative *multiagent tiger problem* [13] ($|S| = 2$, $|A_i| = |A_j| = 3$, $|O_i| = |O_j| = 6$ for level $l \geq 1$, $|O_j| = 3$ at level 0, and $\gamma = 0.9$) *with a total of 5 agents* and 50 models for each other agent. A larger noncooperative *2-agent money laundering (ML) problem* [14] forms the second domain. It exhibits 99 physical states for the subject agent (blue team), 9 actions for blue and 4 for the red team, 11 observations for subject and 4 for the other, with about 100 models

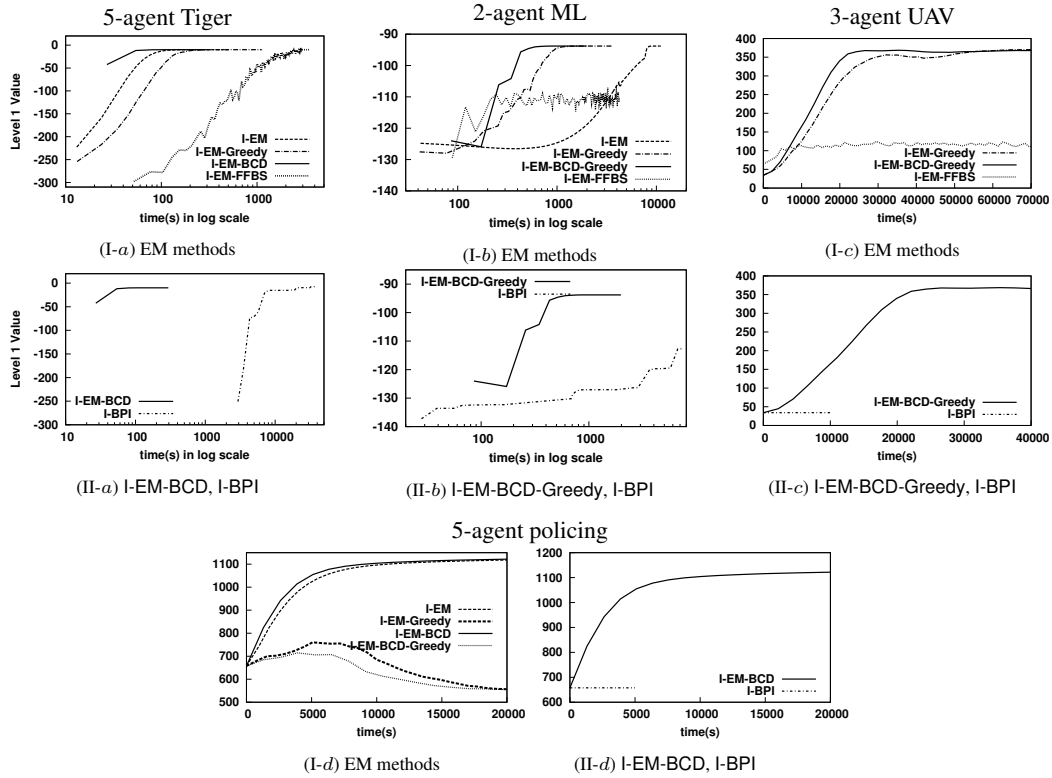

Figure 2: FSCs improve with time for I-POMDP$_{i,1}$ in the (I-$a$) 5-agent tiger, (I-$b$) 2-agent money laundering, (I-$c$) 3-agent UAV, and (I-$d$) 5-agent policing contexts. Observe that BCD causes substantially larger improvements in the initial iterations until we are close to convergence. I-EM-BCD or its greedy variant converges significantly quicker than I-BPI to similar-valued FSCs for all four problem domains as shown in (II-$a$, $b$, $c$ and $d$), respectively. All experiments were run on Linux with Intel Xeon 2.6GHz CPUs and 32GB RAM.

for red team. We also evaluate a *3-agent UAV reconnaissance problem* involving a UAV tasked with intercepting two fugitives in a 3x3 grid before they both reach the safe house [8]. It has 162 states for the UAV, 5 actions, 4 observations for each agent, and 200,400 models for the two fugitives. Finally, the recent *policing protest problem* is used in which police must maintain order in 3 designated protest sites populated by 4 groups of protesters who may be peaceful or disruptive [15]. It exhibits 27 states, 9 policing and 4 protesting actions, 8 observations, and 600 models per protesting group. The latter two domains are historically the largest test problems for self-interested planning.

**Comparative performance of all methods** In Fig. 2-I($a$-$d$), we compare the variants on all problems. Each method starts with a random seed, and the converged value is significantly better than a random FSC for all methods and problems. Increasing the sizes of FSCs gives better values in general but also increases time; using FSCs of sizes 5, 3, 9 and 5, for the 4 domains respectively demonstrated a good balance. We explored various coordinate block configurations eventually settling on 3 equal-sized blocks for both the tiger and ML, 5 blocks for UAV and 2 for policing protest. I-EM and the Greedy and BCD variants clearly exhibit an anytime property on the tiger, UAV and policing problems. The noncooperative ML shows delayed increases because we show the value of agent $i$'s controller and initial improvements in the other agent's controller may maintain or decrease the value of $i$'s controller. This is not surprising due to competition in the problem. Nevertheless, after a small delay the values improve steadily which is desirable.

I-EM-BCD consistently improves on I-EM and is often the fastest: the corresponding value improves by large steps initially (fast non-asymptotic rate of convergence). In the context of ML and UAV, I-EM-BCD-Greedy shows substantive improvements leading to controllers with much improved values compared to other approaches. Despite a low sample size of about 1,000 for the problems, I-EM-FFBS obtains FSCs whose values improve in general for tiger and ML, though slowly and not always to the level of others. This is because the EM gets caught in a worse local optima due

to sampling approximation – this strongly impacts the UAV problem; more samples did not escape these optima. However, forward filtering only (as used in Wu et al. [6]) required a much larger sample size to reach these levels. FFBS did not improve the controller in the fourth domain.

**Characterization of local optima**   While an exact solution for the smaller tiger problem with 5 agents (or the larger problems) could not be obtained for comparison, I-EM climbs to the optimal value of 8.51 for the downscaled 2-agent version (not shown in Fig. 2). In comparison, BPI does not get past the local optima of -10 using an identical-sized controller – corresponding controller predominantly contains listening actions – relying on adding nodes to eventually reach optimum. While we are unaware of any general technique to escape local convergence in EM, I-EM can reach the global optimum given an appropriate seed. This may not be a coincidence: the I-POMDP value function space exhibits a single fixed point – the global optimum – which in the context of Proposition 1 makes the likelihood function, $Q(\pi'_{i,l}|\pi_{i,l})$, unimodal (if $\pi_{i,l}$ is appropriately sized as we do not have a principled way of adding nodes). If $Q(\pi'_{i,l}|\pi_{i,l})$ is continuously differentiable for the domain on hand, Corollary 1 in Wu [16] indicates that $\pi_{i,l}$ will converge to the unique maximizer.

**Improvement on I-BPI**   We compare the quickest of the I-EM variants with previous best algorithm, I-BPI (Figs. 2-II($a$-$d$)), allowing the latter to escape local optima as well by adding nodes. Observe that FSCs improved using I-EM-BCD converge to values similar to those of I-BPI almost *two orders of magnitude faster*. Beginning with 5 nodes, I-BPI adds 4 more nodes to obtain the same level of value as EM for the tiger problem. For money laundering, I-EM-BCD-Greedy converges to controllers whose value is at least 1.5 times better than I-BPI's given the same amount of allocated time and less nodes. I-BPI failed to improve the seed controller and could not escape for the UAV and policing protest problems. *To summarize, this makes* I-EM *variants with emphasis on BCD the fastest iterative approaches for infinite-horizon* I-POMDPs *currently.*

## 6   Concluding Remarks

The EM formulation of Section 3 builds on the EM for POMDP and differs drastically from the E- and M-steps for the cooperative DEC-POMDP [9]. The differences reflect how I-POMDPs build on POMDPs and differ from DEC-POMDPs. These begin with the structure of the DBNs where the DBN for I-POMDP$_{i,1}$ in Fig. 1 adds to the DBN for POMDP hexagonal model nodes that contain candidate models; chance nodes for action; and model update edges for each other agent at each time step. This differs from the DBN for DEC-POMDP, which adds controller nodes for all agents and a joint observation chance node. The I-POMDP DBN contains controller nodes for the subject agent only, and each model node collapses into an efficient distribution on running EM at level 0.

In domains where the joint reward function may be decomposed into factors encompassing subsets of agents, ND-POMDPs allow the value function to be factorized as well. Kumar et al. [17] exploit this structure by simply decomposing the whole DBN mixture into a mixture for each factor and iterating over the factors. Interestingly, the M-step may be performed individually for each agent and this approach scales beyond two agents. We exploit both graphical and problem structures to speed up and scale in a way that is contextual to I-POMDPs. BCD also decomposes the DBN mixture into equal blocks of horizons. While it has been applied in other areas [18, 19], these applications do not transfer to planning. Additionally, problem structure is considered by using FFBS that exploits information in the transition and observation distributions of the subject agent. FFBS could be viewed as a tenuous example of Monte Carlo EM, which is a broad category and also includes the forward sampling utilized by Wu et al. [6] for DEC-POMDPs. However, fundamental differences exist between the two: forward sampling may be run in simulation and does not require the transition and observation functions. Indeed, Wu et al. utilize it in a model free setting. FFBS is model based utilizing exact forward messages in the backward sampling phase. This reduces the accumulation of sampling errors over many time steps in extended DBNs, which otherwise afflicts forward sampling.

The advance in this paper for self-interested multiagent planning has wider relevance to areas such as game play and ad hoc teams where agents model other agents. Developments in online EM for hidden Markov models [20] provide an interesting avenue to utilize inference for *online* planning.

**Acknowledgments**

This research is supported in part by a NSF CAREER grant, IIS-0845036, and a grant from ONR, N000141310870. We thank Akshat Kumar for feedback that led to improvements in the paper.

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
