[Supplementary Material]

## Appendix: Proofs of Propositions

**Proposition 1** (Correctness) The likelihood maximization problem as defined in Eq. 1 with the mixture models as given in Fig. 1 is equivalent to the problem of solving the original I-POMDP$_{i,l}$ of discounted infinite horizon whose solution assumes the form of a finite state controller.

*Proof.* Let $z_i^{0:T} = \{s^t, a_i^t, m_{-i,l-1}, a_{-i}^t, o_i^t, o_{-i}^t\}_0^T$, where $o_i^0$ and $o_{-i}^0$ are null. Equation 1 maximizes the following likelihood:

$$
\begin{aligned}
L(\pi_{i,l}) &= (1-\gamma)\sum_{T=0}^{\infty} \gamma^T Pr(r_i^T = 1|T; \pi_{i,l})\\
&= (1-\gamma)\sum_{T=0}^{\infty} \gamma^T \sum_{z_i^{0:T}} Pr(r_i^T = 1, z_i^{0:T}|T; \pi_{i,l})\\
&= (1-\gamma)\sum_{T=0}^{\infty} \gamma^T \sum_{z_i^{0:T}} Pr(r_i^T = 1|z_i^{0:T}, T; \pi_{i,l})\, Pr(z_i^{0:T}|T; \pi_{i,l})\\
&\propto \sum_{T=0}^{\infty} \gamma^T \sum_{z_i^{0:T}} \gamma^T R_i(s^T, a_i^T, a_{-i}^T)\, Pr(z_i^{0:T}|T; \pi_{i,l})\\
&\propto \sum_{T=0}^{\infty} \gamma^T E_{z_i^{0:T}}\left[R_i(s^T, a_i^T, a_{-i}^T)|\pi_{i,l}\right]
\end{aligned}
$$

The infinite-horizon value function for I-POMDP$_{i,l}$ given a FSC, $\pi_{i,l}$, is:

$$
V_{\pi_{i,l}}(\theta_{i,l}) = \rho(b_{i,l}, a_i^t) + \gamma \sum_{o_i^t} Pr(o_i^t|b_{i,l}^t, a_i^t)\, V_{\pi_{i,l}}(\theta_{i,l}') = \sum_{is} b_{i,l}(is)\, \alpha_{\pi_{i,l}}(is)
$$

where $is = \langle s, m_{-i,l-1}\rangle$, and

$$
\begin{aligned}
\alpha_{\pi_{i,l}}(is) &= E_{a_i, a_{-i}}\left[R_i(s, a_i, a_{-i})|\pi_{,l}\right] + \gamma E_{s, a_i, m_{-i,l-1}, a_{-i}, o_{-i}, o_i}\left[\alpha_{\pi_{i,l}}(is)|\pi_{i,l}\right]\\
&= E_{a_i^0, a_{-i}^0}\left[R_i(s^0, a_i^0, a_{-i}^0)\right] + \sum_{T=1}^{\infty} \gamma^T E_{s^{1:T}, a_i^{1:T}, m_{-i,l-1}^{1:T}, a_{-i}^{1:T}, o_{-i}^{1:T}, o_i^{1:T}}\left[R_i(s^T, a_i^T, a_{-i}^T)|\pi_{i,l}\right]\\
&= \sum_{T=0}^{\infty} \gamma^T E_{s^{1:T}, a_i^{0:T}, m_{-i,l-1}^{1:T}, a_{-i}^{0:T}, o_{-i}^{0:T}, o_i^{0:T}}\left[R_i(s^T, a_i^T, a_{-i}^T)|\pi_{i,l}\right]
\end{aligned}
$$

Subsequently,

$$
\begin{aligned}
V_{\pi_{i,l}(\theta_{i,l})} &= \sum_{is} b_{i,l}(is)\left(\sum_{T=0}^{\infty} \gamma^T E_{s^{1:T}, a_i^{0:T}, m_{-i,l-1}^{1:T}, a_{-i}^{0:T}, o_{-i}^{0:T}, o_i^{0:T}}\left[R_i(s^T, a_i^T, a_{-i}^T)|\pi_{i,l}\right]\right)\\
&= \sum_{T=0}^{\infty} \gamma^T E_{s^{0:T}, a_i^{0:T}, m_{-i,l-1}^{0:T}, a_{-i}^{0:T}, o_{-i}^{0:T}, o_i^{0:T}}\left[R_i(s^T, a_i^T, a_{-i}^T)|\pi_{i,l}\right]\\
&= \sum_{T=0}^{\infty} \gamma^T E_{z_i^{0:T}}\left[R_i(s^T, a_i^T, a_{-i}^T)|\pi_{i,l}\right]
\end{aligned}
$$

$\square$

Consequently, value function of I-POMDP$_{i,l}$ is proportional to the likelihood, and maximizing the latter is equivalent to finding the policy with the optimal value function.

**Propostion 2** (Sufficiency) Distributions, $Pr(a_j^t|s^t)$ across actions, $a_j^t \in A_j$, for each state $s^t$ is sufficient predictive information about the other agent $j$ over all time steps in order to infer the most likely policy of agent $i$. Here,
$Pr(a_j^0|s^0) = \sum_{m_{j,0}^0} Pr(a_j^0|m_{j,0}^0)\, b_{i,1}(m_{j,0}|s^0)$, and
$Pr(a_j^t|s^t) = \sum_{m_{j,0}^t, o_j^t} Pr(m_{j,0}^t|a_j^{t-1}, o_j^t, m_{j,0}^{t-1})O_j(s^t, a_j^{t-1}, o_j^t)\, Pr(a_j^t|m_{j,0}^t)$

*Proof.* Let $z_i^{0:T}=\{s^t, o_i^t, n_{i,l}^t, a_i^t, m_{j,l-1}^t, o_j^t, a_j^t\}$ be a trajectory consisting of the latent state, agent $i$'s FSC node, $j$'s hidden model, and both agents' actions and observations. We use it to expand on the likelihood maximization as given in Eq. 1:

$$
\begin{aligned}
\pi_{i,1}^* &= \arg\max_{\pi_{i,1}} (1-\gamma)\sum_{T=0}^{\infty} \gamma^T\, Pr(r_i^T = 1|T; \pi_{i,1})\\
&= \arg\max_{\pi_{i,1}} \sum_{T=0}^{\infty} \sum_{z_i^{0:T}} Pr(r_i^T = 1, z_i^{0:T}|T; \pi_{i,1})
\end{aligned}
$$

For simplicity of notation, we focus on $l = 1$ and note that the proof holds inductively for any level, $l \geq 1$. Next, we expand the term, $Pr(r_i^T = 1, z_i^{0:T}|T; \pi_{i,1})$:

$$\sum_{z_i^{0:T}} Pr(r_i^T = 1, z_i^{0:T}|T; \pi_{i,1}) = \sum_{n_{i,l}^0} \mathcal{V}_i(n_{i,1}^0) \sum_{s^0} b_{i,1}^0(s^0) \sum_{m_{j,0}^0} b_{i,1}^0(m_{j,0}^0|s^0)$$

$$\times \sum_{a_i^0} \mathcal{L}_i(n_{i,1}^0, a_i^0) \sum_{a_j^0} Pr(a_j^0|m_{j,0}^0) \prod_{t=1}^{T} \sum_{s^t} T_i(s^{t-1}, a_i^{t-1}, a_j^{t-1}, s^t)$$

$$\times \sum_{o_i^t} O_i(s^t, a_i^{t-1}, a_j^{t-1}, o_i^t) \sum_{n_{i,1}^t} \mathcal{T}_i(n_{i,1}^{t-1}, a_i^{t-1}, o_i^t, n_{i,1}^t) \sum_{a_i^t} \mathcal{L}_i(n_{i,1}^t, a_i^t) \sum_{m_{j,0}^t, o_j^t} Pr(m_{j,0}^t|$$

$$a_j^{t-1}, o_j^t, m_{j,0}^{t-1}) O_j(s^t, a_j^{t-1}, o_j^t) \sum_{a_j^t} Pr(a_j^t|m_{j,0}^t) Pr(r_i^T = 1|a_i^T, a_j^T, s^T)$$

Grouping all terms related to each agent, we get:

$$\sum_{z_i^{0:T}} Pr(r_i^T = 1, z_i^{0:T}|T; \pi_{i,1}) = \sum_{n_{i,l}^0} \mathcal{V}_i(n_{i,1}^0) \sum_{s^0} b_{i,1}^0(s^0) \sum_{a_i^0} \mathcal{L}_i(n_{i,1}^0, a_i^0)$$

$$\times \sum_{m_{j,0}^0} b_{i,1}^0(m_{j,0}^0|s^0) \sum_{a_j^0} Pr(a_j^0|m_{j,0}^0) \prod_{t=1}^{T} \sum_{s^t} T_i(s^{t-1}, a_i^{t-1}, a_j^{t-1}, s^t)$$

$$\times \sum_{o_i^t} O_i(s^t, a_i^{t-1}, a_j^{t-1}, o_i^t) \sum_{n_{i,1}^t} \mathcal{T}_i(n_{i,1}^{t-1}, a_i^{t-1}, o_i^t, n_{i,1}^t) \sum_{a_i^t} \mathcal{L}_i(n_{i,1}^t, a_i^t) \sum_{m_{j,0}^t, o_j^t} Pr(m_{j,0}^t|$$

$$a_j^{t-1}, o_j^t, m_{j,0}^{t-1}) O_j(s^t, a_j^{t-1}, o_j^t) \sum_{a_j^t} Pr(a_j^t|m_{j,0}^t) Pr(r_i^T = 1|a_i^T, a_j^T, s^T)$$

$$= \sum_{n_{i,l}^0} \mathcal{V}_i(n_{i,1}^0) \sum_{s^0} b_{i,1}^0(s^0) \sum_{a_i^0} \mathcal{L}_i(n_{i,1}^0, a_i^0) \sum_{a_j^0} Pr(a_j^0|s^0) \prod_{t=1}^{T} \sum_{s^t} T_i(s^{t-1}, a_i^{t-1}, a_j^{t-1}, s^t)$$

$$\times \sum_{o_i^t} O_i(s^t, a_i^{t-1}, a_j^{t-1}, o_i^t) \sum_{n_{i,1}^t} \mathcal{T}_i(n_{i,1}^{t-1}, a_i^{t-1}, o_i^t, n_{i,1}^t) \sum_{a_i^t} \mathcal{L}_i(n_{i,1}^t, a_i^t)$$

$$\times \sum_{m_{j,0}^t, o_j^t} Pr(m_{j,0}^t|a_j^{t-1}, o_j^t, m_{j,0}^{t-1}) O_j(s^t, a_j^{t-1}, o_j^t) \sum_{a_j^t} Pr(a_j^t|m_{j,0}^t) Pr(r_i^T = 1|a_i^T, a_j^T, s^T)$$

$$= \sum_{n_{i,l}^0} \mathcal{V}_i(n_{i,1}^0) \sum_{s^0} b_{i,1}^0(s^0) \sum_{a_i^0} \mathcal{L}_i(n_{i,1}^0, a_i^0) \sum_{a_j^0} Pr(a_j^0|s^0) \prod_{t=1}^{T} \sum_{s^t} T_i(s^{t-1}, a_i^{t-1}, a_j^{t-1}, s^t)$$

$$\times \sum_{o_i^t} O_i(s^t, a_i^{t-1}, a_j^{t-1}, o_i^t) \sum_{n_{i,1}^t} \mathcal{T}_i(n_{i,1}^{t-1}, a_i^{t-1}, o_i^t, n_{i,1}^t) \sum_{a_i^t} \mathcal{L}_i(n_{i,1}^t, a_i^t)$$

$$\times \sum_{a_j^t} Pr(a_j^t|s^t) Pr(r_i^T = 1|a_i^T, a_j^T, s^T)$$

where, $Pr(a_j^0|s^0) = \sum_{m_{j,0}^0} Pr(a_j^0|m_{j,0}^0) b_{i,1}(m_{j,0}|s^0)$, and $Pr(a_j^t|s^t) = \sum_{m_{j,0}^t, o_j^t} Pr(m_{j,0}^t|a_j^{t-1}, o_j^t, m_{j,0}^{t-1}) O_j(s^t, a_j^{t-1}, o_j^t) Pr(a_j^t|m_{j,0}^t)$.

In the last equation above, the only distributions pertaining to $j$ are those over its actions given the state at the initial time step and across time steps up to $T$. $\quad\square$

**Propostion 3** (E-step speed up) Each E-step at level 1 using the forward-backward pass as shown previously results in a net speed up of $\mathcal{O}((|M||\mathcal{N}_{-i,0}|)^{2K}(|\Omega_{-i}|^K))$ over the formulation that ascribes $|M|$ FSCs each to $K$ other agents with each having $|\mathcal{N}_{-i,0}|$ nodes.

*Proof.* In the E-step, we compute $\alpha^t$ and $\beta^h$, which are then used in the M-step. Each of these has complexity, $\mathcal{O}(T_{max}S^2|\mathcal{N}_{i,1}|^2)$, where $T_{max}$ is a bound on $T$ in practice. In order to compute $\widehat{\alpha}$ and $\widehat{\beta}$, we need the transition function of the DBN for given current and next states in the E-step, which has complexity of $\mathcal{O}(S^2|\mathcal{N}_{i,1}^2||A_i||A_{-i}|^K|\Omega_i|)$, where there are $K$ other agents in the environment. E-step's net complexity is given by $\mathcal{O}(S^2|\mathcal{N}_{i,1}|^2(T_{max} + |A_i||A_{-i}|^K|\Omega_i|)$.

A naive formulation infers an FSC for each of $|M|$ level 0 models ascribed to $K$ other agents. Nodes of these controllers are included in the state space of the DBN. The complexity of computing $\widehat{\alpha}$ and $\widehat{\beta}$ is, $\mathcal{O}(T_{max}S^2|\mathcal{N}_{i,1}|^2(|M||\mathcal{N}_{-i,0}|)^{2K})$. In order to compute $\widehat{\alpha}$ and $\widehat{\beta}$, we need the transition function of the Markov model for given current and next states, which has complexity of

$\mathcal{O}(S^2|\mathcal{N}_{i,1}^2|(|M||\mathcal{N}_{-i,0}|)^{2K}|A_i||A_{-i}|^K|\Omega_i||\Omega_{-i}|^K)$. E-step's net complexity for this approach is, $\mathcal{O}(S^2|\mathcal{N}_{i,1}|^2(|M||\mathcal{N}_{-i,0}|)^{2K}(T_{max} + |A_i||A_{-i}|^K|\Omega_i||\Omega_{-i}|^K)$.

The speed up due to our approach is the ratio of the above net complexity of the E-step to the complexity of our E-step:

$$
\begin{aligned}
\mathsf{Speedup} &= \frac{\mathcal{O}(S^2|\mathcal{N}_{i,1}|^2(|M||\mathcal{N}_{-i,0}|)^{2K}(T_{max} + |A_i||A_{-i}|^K|\Omega_i||\Omega_{-i}|^K)}{\mathcal{O}(S^2|\mathcal{N}_{i,1}|^2(T_{max} + |A_i||A_{-i}|^K|\Omega_i|)} \\
&= \frac{\mathcal{O}(S^2|\mathcal{N}_{i,1}|^2(|M||\mathcal{N}_{-i,0}|)^{2K}T_{max})}{\mathcal{O}(S^2|\mathcal{N}_{i,1}|^2(T_{max} + |A_i||A_{-i}|^K|\Omega_i|)} + \\
&\quad \frac{\mathcal{O}(S^2|\mathcal{N}_{i,1}|^2(|M||\mathcal{N}_{-i,0}|)^{2K}(|A_i||A_{-i}|^K|\Omega_i||\Omega_{-i}|^K)}{\mathcal{O}(S^2|\mathcal{N}_{i,1}|^2(T_{max} + |A_i||A_{-i}|^K|\Omega_i|)} \\
&= \mathcal{O}((|M||\mathcal{N}_{-i,0}|)^{2K}) + \mathcal{O}((|M||\mathcal{N}_{-i,0}|)^{2K}|\Omega_{-i}|^K) \\
&= \mathcal{O}((|M||\mathcal{N}_{-i,0}|)^{2K}|\Omega_{-i}|^K)
\end{aligned}
$$

$\square$

**Propostion 4** (E-step ratio at level 0) E-steps in the EMs for obtaining $\phi_{-i,0}$ of $K$ agents exhibits a ratio of complexity, $\mathcal{O}(\frac{|\mathcal{N}_{-i,0}|^2}{|M|})$, compared to the E-steps when $|M|$ FSCs are obtained for $K$ agents.

*Proof.* In the $E$-step presented in this paper, we compute $\widehat{\alpha}$ and $\widehat{\beta}$ (Eq. 4) first. Each of these has complexity, $\mathcal{O}(T_{max}S^2|M|^2)$, where $T_{max}$ is a bound on $T$ in practice. In order to compute $\widehat{\alpha}$ and $\widehat{\beta}$, the transition function of the DBN for given current and next states in the $E$-step has a complexity of $\mathcal{O}(S^2|M|^2|A_{-i}||\Omega_{-i}|)$. E-step's net complexity is then given by $\mathcal{O}(S^2|M|^2(T_{max} + |A_{-i}||\Omega_{-i}|))$. For $K$ other agents, we perform $K$ EMs and the net complexity for $K$ agents is, $\mathcal{O}(S^2|M|^2K(T_{max} + |A_{-i}||\Omega_{-i}|))$.

The naive approach iteratively improves a FSC for each level 0 model. The complexity of computing $\widehat{\alpha}$ and $\widehat{\beta}$ in this case is, $\mathcal{O}(T_{max}S^2|\mathcal{N}_{-i,0}|^2)$. We need the transition function of the DBN for given current and next states to compute $\widehat{\alpha}$ and $\widehat{\beta}$, which has complexity of $\mathcal{O}(S^2|\mathcal{N}_{-i,0}|^2|A_{-i}||\Omega_{-i}|)$. The net complexity of the E-step is given by, $\mathcal{O}(S^2|\mathcal{N}_{-i,0}|^2(T_{max} + |A_{-i}||\Omega_{-i}|)$. For $K$ agents and $|M|$ model each, this becomes, $\mathcal{O}(S^2|\mathcal{N}_{-i,0}|^2|M|K(T_{max} + |A_{-i}||\Omega_{-i}|)$.

The ratio of the complexity of the nai1ve approach to the one presented in this paper is,

$$
\begin{aligned}
\mathsf{Ratio} &= \frac{\mathcal{O}(S^2|\mathcal{N}_{-i,0}|^2|M|K(T_{max} + |A_{-i}||\Omega_{-i}|)}{\mathcal{O}(S^2|M|^2K(T_{max} + |A_{-i}||\Omega_{-i}|))} \\
&= \mathcal{O}(\frac{|\mathcal{N}_{-i,0}|^2}{|M|})
\end{aligned}
$$

This ratio is typically less than 1 because smaller-sized controllers are preferred while the number of models, $|M|$, could get large. $\square$