[Reviews · NeurIPS 2015]

Submitted by Assigned_Reviewer_1

This paper presents an EM method for solving interactive POMDPs (I-POMDPS), which exploits problem structure in the I-POMDP model. Specifically, an EM method for I-POMDPs is introduced, along with improvements which use block-coordinate descent and forward filtering-backward sampling. Experimental results show significant scalability gains using some of these methods.

To the best of my knowledge, this is the first EM method applied to I-POMDPs. While I-POMDPs have many similarities to POMDP (and Dec-POMDPs), where EM has been used, there is additional structure in I-POMDPs in the form of models of the other agents in the problem. As such, while EM could be applied naively to I-POMDPs, more specialized methods could also be developed. Since EM methods have been shown to perform well in problems (particularly when there is significant structure), using EM to solve the very difficult (but structured) I-POMDP problems could be promising.

The proposed EM method exploits the idea that a distribution over other agent actions is sufficient instead of using a distribution over models. This allows more efficient inference, but will still have scalability issues. As a result, the authors include block-coordinate descent, which is an optimization scheme that groups sets of variables and iteratively improves them. Block-coordinate descent could be used in any EM method, but the structure of the I-POMDP may be helpful to setting the blocks.

The authors also introduce a forward filtering-backward sampling (FFBS) method to improve scalability.

The experimental results show significantly improved performance compared to the previous method (bounded policy iteration or BPI) --- larger problems can be solved and other problems can be solved more quickly. Nevertheless, the experiments are missing some combinations of methods making it difficult to analyze the methods. Really, the authors should compare all algorithm combinations on all domains (or give a good reason why this is not possible). The authors also chose the quickest combination of methods *for each problem* to compare against BPI. This is unfair, as you wouldn't know this a priori. Also, FFBS does not perform very well, so the authors should consider additional analysis on this method or removing those results. Additional analysis concerning the experimental results would be helpful. For instance, both BPI and EM can get stuck in local optima, but BPI has the ability to escape some local optima by adding nodes. Why is it that the EM method is able to so significantly outperform BPI?

Also, it is not clear how to set the blocks in block-coordinate descent or how to set the sizes for the other agent finite-state controllers. These are key features affecting the complexity and performance of the methods, so they should be discussed in more detail.

The writing is understandable, but there are several typos and grammatical errors. Additional detail should be given in section 4 (computational complexity) concerning the complexity of the methods themselves. And additional EM-based methods for POMDPs should be discussed such as: H. Li, X. Liao, and L. Carin. Multi-task reinforcement learning in partially observable stochastic environments. Journal of Machine Learning Research, 10:1131-1186, 2009.
Summary: This paper presents the first EM method for I-POMDPs along with two other improvements (based on block-coordinate descent and forward filtering-backward sampling). While some of the proposed methods do not perform well and additional analysis is needed to fully understand the contributions, the methods show promise and some of them consistently outperform the previous state-of-the-art approach.

Submitted by Assigned_Reviewer_2

This paper describes a method for policy search for interactive POMDP s (I-POMDPS) based on several insights to the problem, such as the dependence structure of the model and suitable numerical techniques (bock coordinate descent). The paper specifically focusses on improving the E and M steps of the approach.

Quality: This paper is quite technical but not well motivated. The numerical results are nice and compare the presented technique other approaches on a few standard problems.

Clarity: The language is clear for the most part, but the paper is difficult to follow.

Partially, this is unavoidable in a technical paper, but a better top-level description and transitions between sections would make this paper easier to follow. For example, explaining the relation to other methods more clearly and motivating I-POMDPs rather than simply defining them would make following the paper easier to follow. Terms like "chance node" (L 95) or the notion of "levels" (L55), are stated without describing or motivating them. Do hexagonal nodes behave like round nodes in the DNB? This paper gets too technical too fast and assumes too much familiarity with the particular jargon. One approach to accomplishing this would be to introduce an example early on and use it to motivate both the definitions and to differentiate it from related models such as Dec-POMDP, etc.

Originality:

The current draft does a good job of putting the work in context of similar approaches. While the draft goes into detail about differences in the EM formulation, it does not seem like a drastic departure in terms of approach.

Significance: It is difficult to judge the significance since the paper does not do a good job of placing the work in larger context.

General Comments: This paper would be much stronger if it were more accessible and focused more on the practicality of the approach via numerical examples. Since POMDP are so notoriously difficult to solve giving a solution to a standard problems that were previously intractable in any reasonable amount of time would make this work much more convincing. At this point, the paper is clearly different from other approaches in technical details, but that is not a strong motivation by itself.

Detailed Comments: L65: The term "joint action of all agents" seems like it should be the cross product of all agents. Consider replacing "joint" with "pairwise".

L95: Is there a typo in the inline equation? This term seems to be negative. If the expression denotes proportionality, why include the denominator?

Summary: This paper describes a solution technique for I-POMDP. While there seems to be a contribution the paper would benefit from a better motivation and clearer presentation.

Submitted by Assigned_Reviewer_3

This paper derives a new algorithm for planning in multi agent domains modeled as i-POMDPs. The algorithms derived provide significant speedup over traditional methods and promise to scale better to larger problems. WIth this, this paper addresses an important area of research and presents an approach that is of interest to the research community. The paper is structured and presented well.

There are a number of grammatical mistakes in the paper, including: On page 2, first bullet, the last sentence should be rewritten. Section 3.1, "... about other agent ..." should be "... about other agents ..."
Summary: This paper presents a new algorithm to solve i-POMDP problems for multi-agent planning that is faster than previous ones and promises to allow addressing a wider range of problems. It presents the formalism and shows experimental results illustrating its relative performance.

Author Feedback
Author rebuttal: We appreciate all reviewers' thoughtful comments.

There is strong motivation to study techniques that scale I-POMDPs to larger problems. I-POMDPs offer a general approach for an individual agent to act optimally in partially observable settings shared with other agents who may have conflicting preferences. I-POMDPs do this by maintaining dynamic models and updating both the models and beliefs over them (see [7]). Because of its generality and its perspectivist approach, I-POMDPs are finding applications in domains such as human behavior modeling, improving AI in games, countering money laundering and robot teaching.

Of course, the approach clearly differentiates it from other multiagent frameworks in the space such as Dec-POMDPs, which target the *joint planning problem* for a team of agents (see [7]).

Given page limits and this paper requiring unavoidable technical depth, a balance needed to be struck between presenting more high-level exposition versus ensuring a complete technical description and that the methods can be replicated. We leaned toward the latter hoping that I-POMDPs and their complexity are reasonably known or references can be consulted. We can easily provide more introduction and improve the flow.

#Reviewer1:

We did compare all algorithm combinations in all domains. There are 5 combinations. Solely to preserve clarity and avoid poorly performing methods such as plain I-EM in UAV from crowding out well-performing methods, we do not display some performances. These can be easily brought back into the charts if reviewer wishes.

Indeed, space permitting we would've very much liked to discuss our explorations of the sizes of controllers and sizes of BCD blocks in more detail. Still, we pointed out the following:

- On page7, line365, "Increasing the sizes of FSCs gives better values in
general but also increases time; using FSCs of sizes ... for the 4 domains respectively
demonstrated a good balance." There is nonparameteric work that seeks to find the sizes of the controllers, but this is outside the scope of this particular paper.

- On page7, line367, "We explored various coordinate block configurations eventually settling on 3 equal-sized blocks for both the tiger ... ." Obviously, less blocks lead to fewer but larger optimization subproblems while more blocks lead to smaller subproblems but more of them. This is necessarily a tradeoff that must be evaluated empirically, and there is little guidance on block size by way of theory.

Indeed, the objective of comparing all I-EM variants in Fig2-I is to find out which variant performs uniformly well over 4 domains. From this, our clear recommendation is to utilize I-EM with BCD (greedy and not greedy). Therefore, we compared I-EM with BCD (greedy and not greedy; we show the best performing of the two) with I-BPI.

It surprised us as well to see that I-EM-BCD reached optima that were much better than I-BPI despite allowing the latter to escape. Hence, the paragraph on line382, page8 that notes I-EM can reach the global optima. Another reason is the *peculiar local optima* that confront I-BPI because of the way I-BPI updates alpha vectors. There are many such optima and eventually, escape fails. We will add this explanation on page8.

#Reviewer2:

Motivation for scaling I-POMDPs and how they differ from Dec-POMDPs is given above.

Comparisons of I-EM with I-BPI on 4 domains specifically answer the reviewer's suggestion toward "giving a solution to a standard problems that were previously intractable in any reasonable amount of time ... ." I-BPI is the previous best for infinite-horizon I-POMDPs and the 4 domains are standard problems in this literature. Charts in Fig2-II clearly show that previous best method is inadequate for larger problems. I-EM with BCD scales well to such previously intractable problems. Thus, we present it as the new state of the art for self-interested infinite-horizon planning in multiagent settings.

"chance node" depicted using circle is the usual random variable. This is DBN terminology.

Hexagonal model nodes and edges between them are abstractions. As values of hexagonal nodes are models which are updated between time steps as noted in caption of Fig1; these do not correspond to traditional chance nodes. Fig1(c) shows what's inside successive model nodes and edges between them.

Line 95 has a typo: correct equation is, Pr(r_i^T = 1|a_i^T,a_j^T,s^T) = R_i^T(s^T,a_i^T,a_j^T) - R_{min} / R_{max} - R_{min}. This is the well-known Cooper transformation.

#Reviewer3:

In regards to suggested correction in Section 3.1, the context there is 2 agents i and j. As such, "... about other agent ..." is grammatically correct.

#Reviewer6:

Motivation and significance of the framework and method is given above.

As Reviewer1 and 2 note, the paper clearly differentiates the methods from previous work, other straightforward approaches, and improves on the state of the art.